# Early Diagnosis of Melanoma and Breast Cancer in Women: Influence of Body Image Perception

**DOI:** 10.3390/ijerph19159264

**Published:** 2022-07-28

**Authors:** Jessica Ranieri, Dina Di Giacomo, Federica Guerra, Eleonora Cilli, Alessandra Martelli, Valeria Ciciarelli, Alessandra Ventura, Maria Concetta Fargnoli

**Affiliations:** 1Department of Life, Health and Environmental Sciences (MESVA), University of L’Aquila, P.le S. Tommasi 1, 67010 L’Aquila, Italy; jessica.ranieri@univaq.it (J.R.); dina.digiacomo@cc.univaq.it (D.D.G.); federica.guerra@graduate.univaq.it (F.G.); eleonora.cilli@graduate.univaq.it (E.C.); 2Campus Universitario “Aurelio Saliceti”, University of Teramo, Via Renato Balzarini 1, 64100 Teramo, Italy; amartelli@unite.it; 3UOSD Oncological and General Dermatology, Via Lorenzo Natali 1, 67100 Padova, Italy; valeria.ciciarelli@gmail.com (V.C.); alessandraventura@live.it (A.V.); 4Department of Biotechnological and Applied Clinical Sciences (DISCAB), University of L’Aquila, Via Vetoio, 67100 L’Aquila, Italy

**Keywords:** cancer survivors, well-being, body image, breast cancer, melanoma, early cancer stage

## Abstract

The diagnosis of melanoma and breast cancer may impact many aspects of life with significant reductions in emotional functioning and quality of life. The aim of the study was to analyze the emotional traits of female patients with oncological in early-stage diagnosis, investigating predictors for psychological distress and analyzing body image perception. An observational study was conducted, A sample of 84 female cancer patients (age range 30–55 years) with melanoma (*n* = 42) and breast cancer diagnosis (*n* = 42). The examined emotional variables were psychological distress; depression, stress, and anxiety; metacognitions; and body self-perception. Findings showed higher psychological distress in breast cancer than in melanoma patients (*p* = 0.00), which was related to lower positive self-perception of body image (*p* = 0.03). Furthermore, psychological distress was negatively correlated with consequences of clinical treatment on body image, and low well-being affected the social interaction and well-being with own body. There was no significant difference between cancer staging and timing from diagnosis. Prevention and therapeutic psychological protocols might be adapted and tailored to the unmet needs of the patients in medical treatments to promote and enhance the Quality of Life in survivorship.

## 1. Introduction

Acceleration in psychosocial research efforts over the past 5 years is clear; however, this has not been enough to define psychological aspects and emotional trends of early cancer (melanoma/breast) stage female patients during the clinical trajectory. Currently, new medical treatments are being offered promising long survivorship as a chronic life-threatening disease, however, most of randomized clinical trials are focused on medical treatment outcomes of cancer rather than psychosocial ones. Psychosocial research and practice have failed to keep pace with advances in the medical management of the disease [1,2,3,4,5].

Negative emotional reactions to stressful life events are a normal short-term emotional–behavioral response to threat; anger and fear are not a sign of psychopathological disorder [2,4]. Melanoma and breast cancer may have a considerable impact on patients’ lives including their health-related quality of life. Moreover, psychological responses to diagnosis and treatments may vary substantially over time and according to clinical aspects of the healthcare pathway. Throughout the diagnostic process and survivorship, patients with melanoma/breast cancer report significant reductions in psychological well-being and quality of life, including greater pain and fatigue, insomnia, and greater interference of stressors (physical and emotional) on social activities [1,2,3,4,5,6].

The antecedents and contributing factors to distress and other psychosocial outcomes are largely unexplored. Furthermore, a clear framework for adaptation and effective coping in the context of early-stage cancer is not defined to identify patients at risk of negative psychosocial outcomes in order to structure personalized interventions. Kasparian’s contribution [7] has highlighted oncological patients’ significant psychosocial challenges and consequences of having melanoma. A diagnosis of melanoma may impact all aspects of life. The primary negative impact of melanoma is on an individual personal level (life expectation, body image, self-esteem, well-being); the secondary effect is on social relationships (sociality, work performance, sexuality, family roles).

The psychological distress evoked by a cancer diagnosis, long-lasting threatening survivorship, and outcome uncertainty may impact negatively on patient compliance to medical treatment and win back to everyday life.

Several studies have focused on melanoma and breast cancer linked mental health issues, detecting negative psychological aspects and emotional dimensions. However, a few studies have analyzed the mental health management in early-stage melanoma/breast cancer patients, pointing out body self-perception factors to figure out how the adaptive change is related to oncological therapy, referring to image perception. Assessing body self-perception and identifying its features might serve as a suitable indicator of the cumulative health deterioration that may contribute to a poorer quality of life.

The study aimed to investigate the emotional traits of female patients with early-stage melanoma and breast cancer diagnosis, analyzing body image perception. Breast cancer and melanoma were selected for this study because they have an impact on the physical perception and body image of women, making them relevant for Quality of Life and well-being aspects in survivorship [8,9].

The objective of the study was to analyze the dynamics of psychological dimensions in early-stage breast cancer and melanoma patients by a long follow-up (1–5 years after primary medical treatment); we examined the metacognition thinking, emotional traits, and body self-perception to verify the outcome on well-being.

## 2. Materials and Methods

### 2.1. Ethics Approval

This study was approved by the Institutional Review Board of the University of L’Aquila, Italy (Prot. N° 16,372/2019). Written Informed Consent was obtained from each participant, and all procedures were in accordance with the Declaration of Helsinki.

### 2.2. Participants

Participants were *n* = 84 female patients aged 30–55 years old (mean age 42.2, ±8.3), living in central Italy distributed in 2 groups: (a) Melanoma group (Mg) composed of *n* = 42 patients with melanoma diagnosis (Stage 0-II) and (b) Breast cancer group (Bg) composed of *n* = 42 patients with a breast cancer diagnosis (Stage 0-II). All patients had an early-stage diagnosis by early screening.

Eligible participants have been selected from medical records and approached to be enrolled in the study at S. Salvatore Hospital in L’Aquila (IT) during subsequent scheduled appointments by medical protocol. We contacted 114 eligible patients, of whom 84 provided informed consent; 28 patients did not consent to participate in the experimental protocol; whereas two patients signed the informed consent form, but at the first session, they declined further involvement (dropped out).

Inclusion criteria included: (a) 30–55 years old; (b) no cancer recurrence or second cancer; and (c) timing of disease covering a range of disease 0–120 months from the diagnosis of melanoma/breast cancer; (d) drop-outs long-standing by relevant chemotherapy side effects.

Participants were eligible to enrol in the study if they had a breast cancer diagnosis or melanoma and followed a clinical path within 24 months following diagnosis, and after surgical intervention and/or treatment with adjuvant chemotherapy, radiation therapy, or both, for stages 0-II cancer.

Medical staff applied the TNM classification of Malignant Tumours (Tumor Nodes Metastasis, TNM), a cancer staging system developed by the American Joint Committee on Cancer and the Union for International Cancer Control (UICC), to classify the cancer stage of patients.

The recruitment process is described in Figure 1. Demographic characteristics of the participants are reported in Table 1.

### 2.3. Measurement

#### 2.3.1. Sociodemographic Variables

Sociodemographic data were collected. First, demographic data were provided via the self-report of patients (age, having children, marital status, education, occupation). Second, clinical data were obtained from the patients’ medical records regarding melanoma and breast cancer stage, diagnosis timing, surgical treatments, and pharmacological therapies.

#### 2.3.2. Psychological Tests

The psychological battery was composed of *n* 4 standardized questionnaires aimed to measure psychological distress (Psychological Distress Inventory) to assess signs of anxiety, depression, and stress (Depression Anxiety Stress Scale 21), and to detect metacogni-tive thinking (Metacognition Questionnaire 30).

Psychological Distress Inventory (PDI) [10]: this is a standardized 5-point self-administered 13 item-questionnaire developed to measure the level of psychological distress caused by cancer. The standard score indicates the presence/absence of psychological distress. The internal reliability was good (α = 0.86).

Depression Anxiety Stress Scales-21 (DASS-21) [11]: DASS-21 is a standardized 21-item self-report questionnaire assessing three dimensions (7 items per subscale): depression, anxiety, and stress. Patients are asked to score every item on a scale from 0 (did not apply to me at all) to 3 (applied to me very much). Sum scores are computed by adding up the scores on the items per (sub)scale and multiplying them by a factor of 2. Sum scores for each of the subscales may range between 0 and 42. The internal reliability was good (α = 0.78).

Metacognition Questionnaire-30 (MCQ-30) [12]: it is a standardized self-report assessing a range of metacognitive beliefs and processes relevant to the vulnerability and maintenance of psychological disorders. The items are rated on a 4-point Likert scale from 1 (do not agree) to 4 (completely agree). The items are grouped into five subscales: (a) Cognitive Confidence (CC), (b) Positive Beliefs about Worry (POS), (c) Cognitive Self-Consciousness (CSC), (d) Negative Beliefs about Uncontrollability and Danger (NEG), (e) Need to Control Thoughts (NC). The MCQ-30 has good internal consistency, as do its five subscales. The internal reliability was good (α = 0.80).

Body Self-perception questionnaire (BSPq): It is an ad hoc questionnaire consisting of 16 items aimed to measure three domains of body image perception: (a) consequences of clinical treatment on body image (Treatment Consequences on Body Image, TCBI); (b) well-being in social interactions (Social Wellness, SW), (c) and well-being with your body (Physical Feeling, PF). Responses are based on a 4-point Likert scale. The higher the scores better body self-perception. The pilot study previously conducted showed good reliability. The internal reliability was good (α = 0.82).

### 2.4. Procedure

Medical staff in the Oncological Dermatology Division (Director Prof. M.C. Fargnoli) identified eligible patients. All participants were then enrolled during clinical follow-up according to medical protocol. Written informed consent was obtained at the time of enrolment. Trained clinical psychologists (blinded to the aim of the study) conducted the psychological evaluations, lasting 40 min in a dedicated room. The participants filled out digital versions of the questionnaires; the psychological evaluation lasted 30 min. Participants were recruited on an outpatient basis during their scheduled medical follow-ups. Data were collected anonymously.

### 2.5. Study Design

We conducted an observational study to evaluate the emotional traits, body self-perception, and metacognitions in melanoma and breast cancer patients.

### 2.6. Statistical Analysis

Descriptive statistics were calculated for baseline characteristics and outcome measures. Multivariate analysis of variance (MANOVA) was conducted to detect the statistical significance of the overall differences across the examined psychological variables. Then, Pearson r correlations were applied. Statistical analyses were performed using the Jamovi 1.6.10.0 with a fixed α-value ≤ 0.05.

## 3. Results

Analyzing the sociodemographic data, breast cancer and melanoma patients did not differ significantly by age (*t* = 112, *p* = 0.27), education (Chi2 = 1.64, *p* = 0.44), and marital status (Chi2 = 0.66, *p* = 0.42), or perceived standard of living (Chi2 = 0.92, *p* = 0.63) at 1 month post-diagnosis (T1, see Table 1). Participants resulted homogeneous in demographic data.

In Table 2, we reported clinical treatments. As expected, treatments differed between the types of cancer.

Almost all breast cancer patients were treated with surgery (66.6% lumpectomy and 30.9% mastectomy) and radiotherapy; 38% of them received hormonal therapy, and half of them received chemotherapy. Differently, melanoma patients received surgical excision and no pharmacological therapy.

Both groups (Mg and Bg) showed no signs of anxiety, depression, or stress comparing raw score and cut-off (DASS-21 Anxiety score < 7, DASS-21 Depression score < 9, DASS-21 Stress score < 14). Only Bg showed psychological distress (PDI score > 25). Table 3 reported the raw scores.

Then, MANOVA statistical analyses (3 × 2) were conducted, comparing emotional traits (MCQ, PDI, BPS) and groups (Bg and Mg). Mg evidenced significant positive emotional traits than Bg: Bg showed higher psychological distress (*p* = 0.00), whereas Mg showed better Body Image Perception (*p* = 0.03), in the Impact of Treatment Consequences on Body Image index (*p* = 0.01) and on social wellness index (*p* = 0.04) (see Figure 2).

Further, we examined the relationship between body image perception (BSPq: TCBI, SW, PF) and psychological dimension (PDI), as well as metacognitive beliefs (MCQ-30: POS, NEG, CC, NC, CSC). We conducted a Pearson’s correlation analysis: the results are summarized in Figure 3. Psychological distress (PDI) was negatively correlated with consequences of clinical treatment on body image (TCBI) (r = −0.5; *p* < 0.001), well-being in social interaction (SW) (r = −0.4; *p* < 0.001), and well-being with own body (PF) (r = −0.5; *p* < 0.001). Then, metacognitive beliefs (MCQ-30) were negatively correlated with body image perception (BSPq): significant correlations emerged between TCBI and NEG (r = −0.5; *p* < 0.001), CC (r = −0.4; *p* < 0.001), NC (r =−0.2; *p* = 0.01); between SW and NEG (r = −0.3; *p* = 0.003), CC (r = −0.3; *p* = 0.005), NC (r =−0.2; *p* = 0.02); between PF and NEG (r = −0.2; *p* = 0.001), CC (r = −0.4; *p* < 0.001). Last, timing from diagnosis (time range 0–120 months) resulted in no significance.

## 4. Discussion

The aim of the study was to analyze the emotional traits of female patients with early-stage breast cancer and melanoma diagnosis, verifying the diagnosis impact on body image perception; in particular, the focus of the study was to define the predictive factor of psychological stress by metacognition thinking, emotional traits and body self-perception on the younger sample (range age 30–55 years).

According to the emerging literature on the well-being of younger cancer patients in survivorship [11,12,13,14,15,16], our findings showed that young women with melanoma/breast cancer resilient, as in our sample, showed no signs of mood disorders or psychopathological conditions.

According to Bourdon’s suggestions [17], melanoma patients showed fewer negative physical and social consequences than breast cancer patients, which decreased their body image self-perception as physical and role functioning, experiencing more extensive and painful treatment. Breast cancer patients have higher psychological distress and low body image perception than melanoma patients; it is interesting to highlight the age effect across patient groups, with younger patients seeming to better impact the disease pathway.

Our study shows that early-stage melanoma patients represent an oncological target with a better adaptability and management of their own body in the post-diagnosis than breast cancer patients. Early screening, as an effective tool for the early diagnosis of melanoma, favors not only less invasive clinical treatments but also a greater awareness of one’s body with a better management of it in the post-diagnosis, thus laying down for a better quality of life and well-being during survivorship.

Taking into account breast cancer patients’ performance, the diagnosis represents a stressor not only for illness management but even for physical perception, impacting significantly on the psychological dimensions of body image. Our finding evidenced significant relations of body image perception with emotion regulation and metacognitive beliefs of breast cancer women: low body self-perception appeared related to high psychological distress; moreover, high body self-perception was related to lower levels of unhelpful metacognitions.

Our findings confirmed the investigated largely negative impact of cancer diagnosis [2,5,8,13,14,15,16] and contributed to emerging research topics regarding the clinical implication of early diagnosis at a younger age (in this case in the female population), detecting the high relevance of clinical psychological interventions among medical and pharmacological solutions in primary treatments. While cancer treatment is being planned, all women should be informed about the possible side-effects of treatment on body image. Despite that the psychological impact of cancer on body image in young women has been identified, self-perception issues are not systematically addressed in the comprehensive care to these patients.

Emotional and metacognition screening and body image perception of BC and melanomal patients should be encouraged to prevent and/or reinforce by tailored psychological (integrated) treatment to enhance metacognitive beliefs for a better mantainence of a good level of body image perception.

This study has some limitations. First, this study is an observational study, which is difficult to determine the causal relationships; further follow-up studies are needed. The psychological battery is composed of self-report measures of distress symptoms and metacognitive thinking and are not measures of their clinical indicators. Further, the relatively small sample size limits the generalizability of our findings regarding regaining a normal life after the clinical pathway.

In conclusion, our study evidenced BC women had lower well-being compared to women with melanoma in survivorship. The study suggested that psychological support in primary treatments should be tailored to enhance the recovery of well-being and Quality of Life in survivors. Priority for psychological treatments should be adapted to the unmet needs of the patients in the medical pathway and for their own life perspectives. Last, our finding highlighted the need for regular screening of psychological distress even in patients with an early cancer diagnosis and in the case of long-term follow-up.

## Figures and Tables

**Figure 1 ijerph-19-09264-f001:**
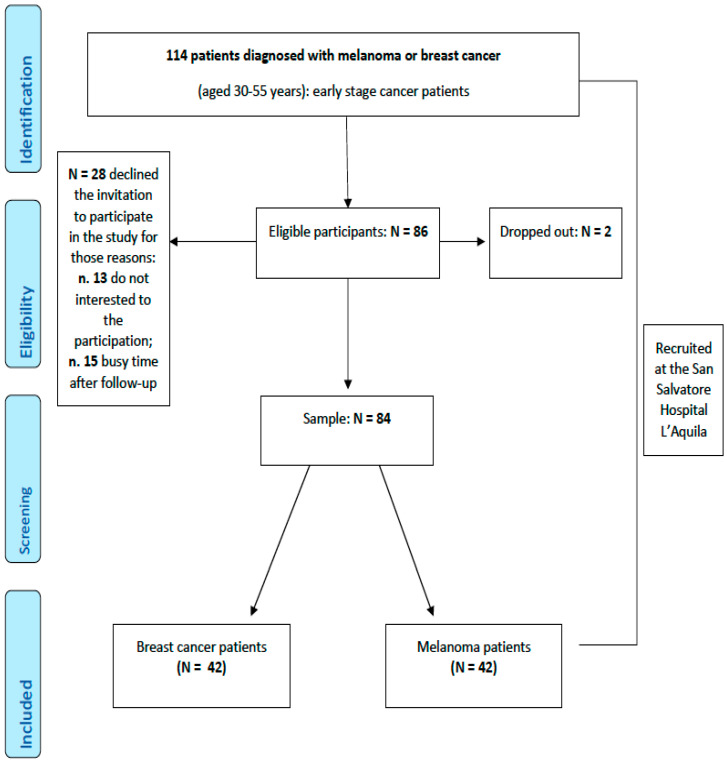
Flowchart of the participants.

**Figure 2 ijerph-19-09264-f002:**
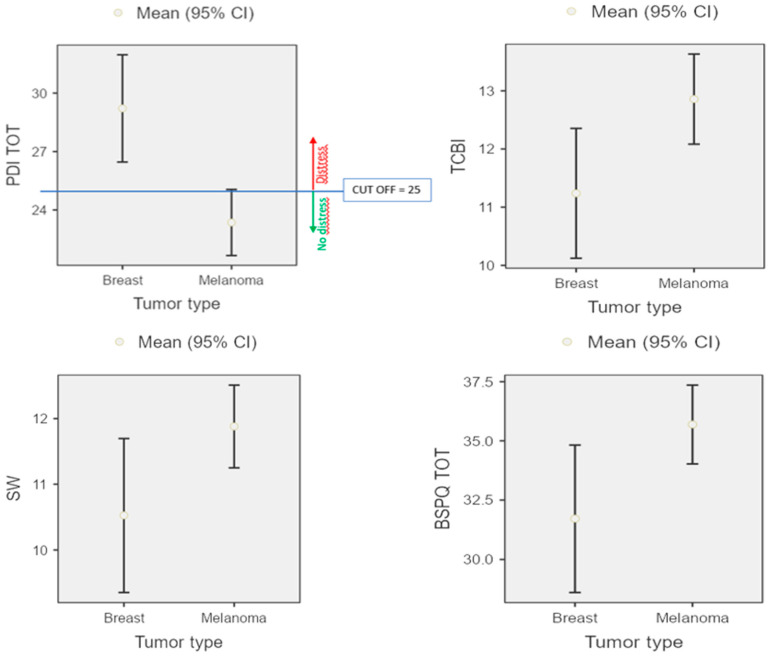
Psychological distress and self-body perception representation of both Bg and Mg groups.

**Figure 3 ijerph-19-09264-f003:**
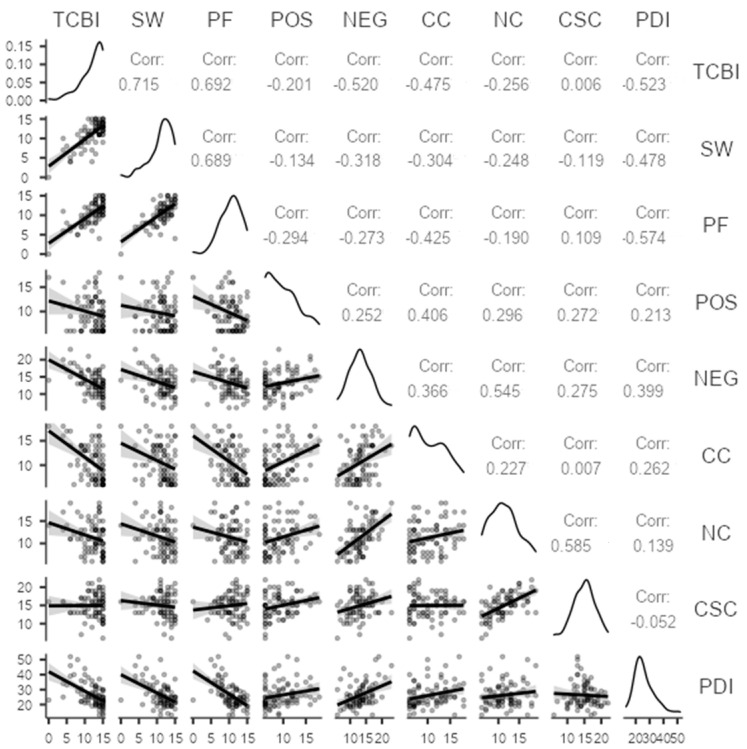
Pearson r correlations between PDI, BSPq, and MCQ-30 scores for the sample.

**Table 1 ijerph-19-09264-t001:** Sociodemographic data of participants.

Age, Years	Breast Group (Mean Age 45.3, Sd ± 1.5)	Melanoma Group (Mean Age 45.5, Sd ± 2.5)
Relationship status		
Married/living with partner	80.9%	78.5%
Single	19.0%	21.4%
Education		
Not graduate	19.0%	4.7%
High school	52.3%	64.2%
Graduation	28.5%	30.9%
Occupation		
Housewife	27.9%	23.8%
Employed	53.5%	57.1%
Self-employed	17.4%	19.0%
TNM Cancer stage		
0	47.6%	40.4%
I	28.5%	57.1%
II	23.8%	2.3%
Diagnosis Timing		
T0 (0–12 months)	30.9%	21.4%
T1 (13–60 months)	57.1%	45.2%
T2 (>61 months)	11.9%	33.3%

**Table 2 ijerph-19-09264-t002:** Clinical treatments for patients with breast cancer and melanoma.

	Melanoma Group	Breast Cancer Group
Surgery		
Excision	(*n* = 42) 100%	-
Lumpectomy	-	(*n* = 29) 69.0%
Mastectomy	-	(*n* = 13) 30.9%
Pharmacological therapy *		
No treatment	(*n* = 42) 100%	(*n* = 3) 7.1%
Chemotherapy	-	(*n* = 22) 52.3%
Hormonal therapy	-	(*n* = 16) 38.0%

* Chemotherapy and hormonal therapy are not mutually excluded.

**Table 3 ijerph-19-09264-t003:** The raw score (mean and standard deviations) of participants in the psychological evaluation.

Test	Breast Cancer Group		Melanoma Group	
	M	Sd	M	Sd
MCQ-30	58.5	10.3	60.7	12.4
POS	9.0	3.0	10.2	3.4
NEG	13.1	3.5	13.2	3.3
CC	10.6	3.8	10.4	3.4
NC	11.0	3.4	11.6	3.2
CSC	14.6	3.1	15.2	3.6
PDI	29.2	8.8	23.3	5.4
BSP	31.7	9.9	35.6	5.3
TCBI	11.24	3.5	12.8	2.4
SW	10.5	3.7	11.8	2.0
PF	9.9	3.5	10.9	2.0
DASS-21				
D	4.1	4.2	3.0	2.7
A	4.0	3.5	3.5	3.0
S	6.2	4.4	5.6	2.6

M = mean value; sd = standard deviation.

## Data Availability

Not applicable.

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
