# Peer review of "Early Diagnosis of Melanoma and Breast Cancer in Women: Influence of Body Image Perception"

_ijerph, 2022, doi:10.3390/ijerph19159264_

Round 1
Reviewer 1 Report
The Authors present the paper: "Early diagnosis of melanoma and breast cancer in women: influence of body image perception" reporting a modest study.
1) The size of participants is small
2) It should be interesting to have a long-term follow-up
3) The Authors should report the used questionnaires in Appendix
4) The Authors should specify the time requested to fill-out the questionnaires
5) The Conclusions should report, even if in general, what type of psychological support they suggest to have
6) The study limitations, well reported by Authors, underline its modest content
Author Response
Dear Reviewer
I uploaded the revised paper associated with point-to-point replying
Thanks for your suggestions
My best
Reviewer Comments’ Replying
Title
Early diagnosis of melanoma and breast cancer in women: influence of body image perception
Below the replying point-to-point. All changing were processed by ‘Track Changes’.
Dear Reviewer 1
We are grateful for your suggestions. We tried to process your suggestions improving the manuscript quality. Below we reported our ‘points-to-points’ replying. All changing were processed by ‘Track Changes’.
R1 = Reviewer 1
A = Authors
R1: The Authors present the paper: "Early diagnosis of melanoma and breast cancer in women: influence of body image perception" reporting a modest study.
- The size of participants is small
- It should be interesting to have a long-term follow-up
- The Authors should report the used questionnaires in Appendix
- The Authors should specify the time requested to fill-out the questionnaires
- The Conclusions should report, even if in general, what type of psychological support they suggest to have…
- The study limitations, well reported by Authors, underline its modest content
A: Despite the effort for the Reviewer, we are sorry that our research was under his/her expectations. Regard to single points:
1) Normally, research protocol by psychological approach including oncological patients could be based on several sample size. Our study is in line with literature;
2) thanks for suggestion: we will take care about that in on going study;
3) We are sorry but the psychological tests are under copyright and we aren’t entitled to publish them;
4) We added the completion time of the psychological battery. (Please see line 227)
5) We tried to explain better the psychological support. (Please see lines 369-372)
R1: The study limitations, well reported by Authors, underline its modest content
A: We are sorry for that.
Reviewer 2 Report
Dear Authors,
I read the manuscript with great interest.
In my opinion, the paper needs corrections before publication.
Abstract:
The abstract does not include the most important parts like methodology, results, and conclusions. It is mentioned that higher psychological distress in breast cancer than in melanoma patients …..however there are no results described (expressed in n; %). The methodology should involve a short description of questionnaires used in the study, for the assessment of depression we used standardized questionnaires, etc. Moreover, based on 84 patients you are not allowed to calculate the prevalence. The conclusions do not correspond with the aims of the study.
Could the authors explain why they decided to combine melanoma and breast, cancer patients? Isn’t that breast cancer has a bigger impact on females? It is only said that there is an impact on physical perception.
Material and Methods:
Authors wrote: Participants were n.84 female patients aged 30-55 years old (mean age 42.2, 8.3), 72 living in central Italy distributed in n.2 groups: a) M group (Mg) composed of n. 42 patients with melanoma diagnosis (Stage 0-II) and b) B group (Bg) c
Isn’t it better to change “.” for “=” in n.84 for n=84?
Please rename groups: instead of M group write Melanoma group (Mg).
Results:
Could the authors provide an explanation of the v2 test? is it another name for the Chi2 test?
I don’t understand the purpose of Table 2? it is necessary to add n values, as well.
A better description of statistical methods is needed. There is a difference between ANOVA and MANOVA.
Is the Pearson correlation appropriate here? Did the authors check the normal distribution of psychological factors?
The big advantage of the article is the length of it, good job.
The biggest limitation: no results regarding the prevalence are described. No Frequencies and 95% CI. Please, remove that from your aim of the study or improve.
There are some typos, grammar issues, please, revise.
The references should be prepared according to journal instructions.
Author Response
Dear Reviewer
herewith I uploaded the poin-to-point replying
Thanks for your suggestions.
My Best
Reviewer Comments’ Replying
Title
Early diagnosis of melanoma and breast cancer in women: influence of body image perception
Below the replying point-to-point. All changing were processed by ‘Track Changes’.
Reviewer 2 (R2 = Reviewer 2; A = Authors)
R2: Dear Authors, I read the manuscript with great interest. In my opinion, the paper needs corrections before publication.
Abstract:
The abstract does not include the most important parts like methodology, results, and conclusions. It is mentioned that higher psychological distress in breast cancer than in melanoma patients …..however there are no results described (expressed in n; %).
A: We tried to improve the abstract sections. Thanks.
R2: The methodology should involve a short description of questionnaires used in the study, for the assessment of depression we used standardized questionnaires, etc.
A: We tried to briefly describe the psychological battery.
R2: Moreover, based on 84 patients you are not allowed to calculate the prevalence.
A: We made correction. Thanks
R2: The conclusions do not correspond with the aims of the study.
A: We tried to improve the conclusion section.
R2: Could the authors explain why they decided to combine melanoma and breast, cancer patients? Isn’t that breast cancer has a bigger impact on females? It is only said that there is an impact on physical perception.
A: We explained our target in introduction paragraph. (please see lines 198-200)
R2: Material and Methods:
Authors wrote: Participants were n.84 female patients aged 30-55 years old (mean age 42.2, ±8.3), 72 living in central Italy distributed in n.2 groups: a) M group (Mg) composed of n. 42 patients with melanoma diagnosis (Stage 0-II) and b) B group (Bg) c
Isn’t it better to change “.” for “=” in n.84 for n=84?
A: Done; we changed “.” for “=”.
R2: Please rename groups: instead of M group write Melanoma group (Mg).
A: Done. Thanks!
R2: Results: Could the authors provide an explanation of the v2 test? is it another name for the Chi2 test?
A: Sorry for the unvolutary mistake: our platform changed the letters. Very sorry!
R2: I don’t understand the purpose of Table 2? it is necessary to add n values, as well.
A: We added n values. Thanks!
R2: A better description of statistical methods is needed. There is a difference between ANOVA and MANOVA.
A: We have corrected it.
R2: Is the Pearson correlation appropriate here? Did the authors check the normal distribution of psychological factors?
A: We checked and the calculation was correct.
R2: The big advantage of the article is the length of it, good job.
A: We are grateful for your feedback.
R2: The biggest limitation: no results regarding the prevalence are described. No Frequencies and 95% CI. Please, remove that from your aim of the study or improve.
A: We did it. Thanks (Please see line 13).
R2: There are some typos, grammar issues, please, revise.
A: Done.
R2: The references should be prepared according to journal instructions.
A: Done.
Reviewer 3 Report
The authors studied the emotional traits of female patients with early-stage breast cancer and melanoma, verified the diagnosis's impact on body image perception, and tried to define the predictive factor of psychological stress by metacognition thinking and body-self-perception in the younger sample. The review highlights the importance of early screening as an effective tool for the early diagnosis of melanoma and breast cancer, and also suggests better ways to improve patients’ quality of life.
Comments:
1. Smoking and drinking are associated with a significant increase in breast cancer and melanoma risk. The study used MCQ-30 Questionnaire in the younger population, did the authors find any impact of these habits on the disease progression and psychological stress levels? Any correlation between them?
2. Typos and grammar checks are required
Author Response
Dear Reviewer
I uploaded the revised paper associated with point-to-point replying
Thanks for your suggestions
My best
Reviewer Comments’ Replying
Title
Early diagnosis of melanoma and breast cancer in women: influence of body image perception
Below the replying point-to-point. All changing were processed by ‘Track Changes’.
Reviewer 3. (R3 = Reviewer 3; A = Authors)
R3: The authors studied the emotional traits of female patients with early-stage breast cancer and melanoma, verified the diagnosis's impact on body image perception, and tried to define the predictive factor of psychological stress by metacognition thinking and body-self-perception in the younger sample. The review highlights the importance of early screening as an effective tool for the early diagnosis of melanoma and breast cancer, and also suggests better ways to improve patients’ quality of life.
Comments:
- Smoking and drinking are associated with a significant increase in breast cancer and melanoma risk. The study used MCQ-30 Questionnaire in the younger population, did the authors find any impact of these habits on the disease progression and psychological stress levels? Any correlation between them?
A: We didn’t detect lifestyle and health behaviors variables. However, we believe that the investigation of self-care is an interesting starting point to be explored in future studies and an emerging topic in health psychology. Thank you for your suggestion.
R3: 2. Typos and grammar checks are required
A: Done.
Reviewer 4 Report
The paper describes the emotional traits of female patients with oncological diagnosis (early stage melanoma/breast cancer) and investigates the prevalence and predictors for psychological stress analyzing body image perception. Findings showed higher psychological distress in breast cancer than melanoma patients, which was related to lower positive self-perception of body image.
I report below some suggestions to improve the quality of the article.
Abstract
The sentence “diagnosis of melanoma/breast cancer stage T0-T2” could be not clear to audience with low expertise on the topic, please provide an explanation of T0-T2 or remove this information.
Introduction
Line 58: please correct the sentence in: “This study aimed to investigate the emotional traits of female patients with early stage 58 melanoma and breast cancer diagnosis analyzing body image perception”.
Line 61: it sounds better to report “we aimed to” rather than “we wanted to”.
Methods
Participants
-How the researchers managed recruitment? Were the 114 eligible patients contacted selected from a specific database or medical records? Which years for considered for study inclusions?
-could the Authors provide more details on the following inclusion criterion “no cancer recurrence (2nd primary 82 diagnosis”?
-this sentence is not clear “diagnosis of melanoma/breast cancer had been in the time period from 0 to 120 months”
-Line 88: the acronym TMN should be explained
-How do the Authors think to manage eventual drop-outs or change in expected timeline, which could occur also in case of remote intervention delivery due to fatigue and possible side effects of chemotherapy?
Table 1
-Are there any statistical differences between variables distribution in the two groups? Could the Authors report in the Table the p value for between group comparisons?
Measurement
-The Authors should report which demographic and clinical data have been reported for each patient.
-psychometric properties are reported only for Psychological Distress Inventory. However, the same information for the different questionnaires should be provided.
-A Statistical Analysis paragraph should be inserted
Results
-Why in table 3 did the Authors report χ in the first column of each condition as mean (breast cancer vs melanoma)? Should it be M?
Discussion
-How do study data can sustain this sentence “Our study shows that early stage melanoma patients represent an oncological target 214 with the best adaptability and management of their own body in the post-diagnosis”?
-Is the sentence “low body self-perception appeared predictive for high psychological distress” related to data of correlations? In this case, please correct, as correlation analyses could not lead to interpretation of causal relationship between variables.
Author Response
Dear Reviewer
I uploaded the revised paper associated with point-to-point replying
Thanks for your suggestions
My best
Reviewer Comments’ Replying
Title
Early diagnosis of melanoma and breast cancer in women: influence of body image perception
Below the replying point-to-point. All changing were processed by ‘Track Changes’.
Reviewer 4. (R4 = Reviewer 4; A = Authors)
R4: The paper describes the emotional traits of female patients with oncological diagnosis (early stage melanoma/breast cancer) and investigates the prevalence and predictors for psychological stress analyzing body image perception. Findings showed higher psychological distress in breast cancer than melanoma patients, which was related to lower positive self-perception of body image.
I report below some suggestions to improve the quality of the article.
Abstract
The sentence “diagnosis of melanoma/breast cancer stage T0-T2” could be not clear to audience with low expertise on the topic, please provide an explanation of T0-T2 or remove this information.
A: We removed “T0-T2” and added “early stage” in line 3.
R4: Introduction
Line 58: please correct the sentence in: “This study aimed to investigate the emotional traits of female patients with early stage 58 melanoma and breast cancer diagnosis analyzing body image perception”.
A: Done (please see lines 64-65).
R4: Line 61: it sounds better to report “we aimed to” rather than “we wanted to”.
A: We rephrased the sentence (please see lines 74-75).
R4: Methods
Participants
-How the researchers managed recruitment? Were the 114 eligible patients contacted selected from a specific database or medical records? Which years for considered for study inclusions?
A: We improved recruitment and eligibility information in lines 91-93.
R4: -could the Authors provide more details on the following inclusion criterion “no cancer recurrence (2nd primary 82 diagnosis”?
A: We improved the part. (Please see lines 97-98)
R4: -this sentence is not clear “diagnosis of melanoma/breast cancer had been in the time period from 0 to 120 months”
A: We improved the sentence in lines 99-100.
R4: -Line 88: the acronym TMN should be explained
A: Done (please see lines 104-105).
R4: -How do the Authors think to manage eventual drop-outs or change in expected timeline, which could occur also in case of remote intervention delivery due to fatigue and possible side effects of chemotherapy?
A: Those key points were inserted in exclusion criteria. (Please see line 218)
R4: Table 1
-Are there any statistical differences between variables distribution in the two groups? Could the Authors report in the Table the p value for between group comparisons?
A: We conducted the Shapiro test and no differences emerged
R4: Measurement
-The Authors should report which demographic and clinical data have been reported for each patient.
A: Done.
R4: -psychometric properties are reported only for Psychological Distress Inventory. However, the same information for the different questionnaires should be provided.
A: Done!
R4: -A Statistical Analysis paragraph should be inserted
A: Done.
R4: Results
-Why in table 3 did the Authors report χ in the first column of each condition as mean (breast cancer vs melanoma)? Should it be M?
A: We converted “x” to “M”.
R4: Discussion
-How do study data can sustain this sentence “Our study shows that early stage melanoma patients represent an oncological target 214 with the best adaptability and management of their own body in the post-diagnosis”?
A: We tried to improve the sentence. Please see lines 239-240.
R4: -Is the sentence “low body self-perception appeared predictive for high psychological distress” related to data of correlations? In this case, please correct, as correlation analyses could not lead to interpretation of causal relationship between variables.
A: We adapted the part. Please see line 248.
Round 2
Reviewer 1 Report
The Authors reviewed the manuscript completing the requests of the reviewers.
At this point I consider the modifications and adds appropriate making the manuscript more complete and understandable
Author Response
Dear Reviewer
we are grateful for your appretiation.
Best regards
Dina Di Giacomo
Reviewer 2 Report
Dear Authors,
please, provide final corrections.
Alessandra Martelli, add a specific affiliation. Department of .... etc.
There are differences in the size of the font in the abstract.
Last sentence in the abstract:
Prevention and therapeutic psycholigcal protocols might 21 be adapted and tailored to the unmet need of the patients in medical treatmens to promote and enhance the Quality of Life quality in survivorship. Isn't the term quality doubled here?
references which are continous 1,2,3,4,5 should be described as 1-5
I don't know why quality of life is with capital letters.
Sometimes there is a space between Chi and 2 sometimes there is not.
Good luck!
Author Response
Dear Reviewer
we worked on all your suggestions as you can see by change tracking on main text.
The point related to Quality of Life: it is an indicator for well-being and normally is reported in Capitol letters.
Many thansk
Dina Di Giacoomo